# Additional Value of Patient-Reported Symptom Monitoring in Cancer Care: A Systematic Review of the Literature

**DOI:** 10.3390/cancers13184615

**Published:** 2021-09-15

**Authors:** Luís Lizán, Lucía Pérez-Carbonell, Marta Comellas

**Affiliations:** 1Department of Medicine, Jaume I University, 12071 Castellón de la Plana, Spain; 2Outcomes’10, Jaume I University, 12071 Castellón de la Plana, Spain; lperez@outcomes10.com (L.P.-C.); mcomellas@outcomes10.com (M.C.)

**Keywords:** patient-reported symptoms, survival, Health-Related Quality of Life, satisfaction, use of resources

## Abstract

**Simple Summary:**

The additional value of patient-reported symptom monitoring in routine cancer is still under discussion. With this in mind, we have reviewed recent evidence on the benefits of this strategy. The evidence examined illustrates that bringing systematic patient feedback into the oncology consultation provides objective advantages over usual care, such as better symptom control, early detection of tumor recurrence, and extended chemotherapy use. Such care improvements ultimately entail an outstanding survival benefit for advanced cancer patients, an increase in their global quality of life, and eventually, medical cost savings. Monitoring patient-reported symptoms might also have other implications in clinical practice, such as promoting patient disease awareness or enhancing patient–physician communication and relationships. Notwithstanding these advantages, there are still logistical barriers that prevent its widespread implementation—especially in the electronic modality. In addition, the real-world effectiveness and the cost-effectiveness of this strategy are yet to be proven in different settings.

**Abstract:**

Background: To describe the benefit of patient-reported symptom monitoring on clinical, other patient-reported, and economic outcomes. Methods: We conducted a systematic literature review using Medline/PubMed, limited to original articles published between 2011 and 2021 in English and Spanish, and focused on the benefit of patient-reported symptom monitoring on cancer patients. Results: We identified 16 reports that deal with the benefit of patient-reported symptom monitoring (collected mostly electronically) on different outcomes. Five studies showed that patient-reported symptom surveillance led to significantly improved survival compared with usual care—mainly through better symptom control, early detection of tumor recurrence, and extended chemotherapy use. Additionally, three evaluations demonstrated an improvement in Health-Related Quality of Life (HRQoL) associated with this monitoring strategy, specifically by reducing symptom severity. Additionally, six studies observed that this monitoring approach prevented unplanned emergency room visits and hospital readmissions, leading to a substantial decrease in healthcare usage. Conclusions: There is consistent evidence across the studies that patient-reported symptom monitoring might entail a substantial survival benefit for cancer patients, better HRQoL, and a considerable decrease in healthcare usage. Nonetheless, more studies should be conducted to demonstrate their effectiveness in addition to their cost-effectiveness in clinical practice.

## 1. Introduction

Cancer patients suffer from significant physical and psychosocial consequences derived from either the disease itself or cancer treatment toxicities [1]. For years, physicians have mainly focused on traditional oncological outcomes such as overall or disease-free survival, and tumor response, to evaluate and monitor cancer patients. In contrast, they might have overlooked cancer-related symptoms such as pain, nausea, or headaches [2,3,4], or other consequences of cancer such as distress and fatigue [5]. Underrecognizing and undertreating cancer-related symptoms might directly affect the continuity of cancer treatment. In addition, it might increase cancer morbidity or the need for healthcare resources, thus adding substantial medical costs [6]. 

Information about physical symptoms such as pain, nausea, and psychological consequences (anxiety, depression, or sleep disturbances) can only be obtained directly from patients. Thus, the use of the Patient-Reported Outcomes (PROs) assessment during cancer patients’ follow-up may provide a more complete picture of their general health. In addition, the development of technology might facilitate the collection of these PROs through computers, tablets, or smartphones so that patients can report their symptoms during and between visits. Moreover, at the same time, healthcare providers can receive real-time alerts in case of clinical deterioration, whereby they can respond rapidly [7]. According to different reviews, gathering patient-reported symptoms (including physical symptoms and psychological disturbances) during cancer patients’ follow-up may lead to an improvement in patient management and symptom control [8,9], may improve patient satisfaction with treatment [9,10], and may improve the communication between clinicians and their patients [10,11]. 

Despite the growing interest in patient-reported symptom monitoring, especially in the electronic modality, there is still much debate as to its additional value in routine cancer follow-up. Indeed, there is insufficient understanding of the impact of patient-reported symptom monitoring on health outcomes [10,11,12]. Therefore, this systematic review aimed to describe the benefits of patient-reported symptom monitoring—regardless of the modality (paper-based or electronic)—on health outcomes such as clinical (e.g., survival), patient-reported (e.g., Health-Related Quality of Life [HRQoL], general perception or feelings of well-being, satisfaction, etc.) and economic outcomes (use of healthcare resources, costs, cost-effectiveness, etc.). 

## 2. Materials and Methods

### 2.1. Search Design 

We conducted a systematic review of the literature according to the Cochrane methodology [13] and the Preferred Reporting Items for Systematic Reviews and Meta-Analyses (PRISMA) checklist for reporting [14] (see Appendix A). The protocol for our systematic review was registered in the search registry database (reviewregistry1221). We acknowledge that a large amount of evidence regarding PROs and cancer has emerged in recent years; thus, we developed a search strategy to maximize the identification of recent literature about the benefits of patient-reported symptom monitoring as an intervention, but we tried to find the best balance between sensitivity and specificity for retrieving publications. To do so, we searched the terms related to the population (cancer) and the intervention (PROs and patient-reported symptoms) as free-text keywords in the title field. Then, we narrowed the search adding terms related to the outcomes, using either free-text or MeSH terms when available. Additionally, we avoided overlapping terms, that is, those that do not add new results to the search. Terms were combined with Boolean operators (AND/OR), and the search was adapted to the Medline/PubMed international database. Appendix A show the research strategy in more details. Additionally, we manually searched the reference lists of relevant original articles and reviews obtained in the search. We also carried out a search of the grey literature using the search engine Google Scholar and combined text terms such as patient reported-symptoms, survival, satisfaction, adherence, resources, and costs. Furthermore, we reviewed international congress pages related to outcome research and pharmacoeconomics, such as the International Society for Pharmacoeconomics and Outcomes Research (ISPOR). 

The search was limited to original articles published in the last ten years (2011–2021) in English or Spanish, conducted in Europe, North America, and Australia, focusing on (1) the use of patient-reported symptom monitoring in the context of clinical trials or routine practice in cancer patients, and (2) the impact of patient-reported symptom monitoring on health outcomes, including clinical (e.g., survival), patient-reported (e.g., HRQoL, general perception or feelings of well-being, satisfaction, etc.) or economic outcomes (use of healthcare resources, costs, cost-effectiveness, etc.).

### 2.2. Data Abstraction and Quality Assessment

Two researchers independently screened each of the identified publications based on their titles, abstracts, and full texts for inclusion criteria. Any discrepancies between reviewers were resolved through consensus and, if necessary, by consulting a third reviewer. Table 1 shows the inclusion criteria following the PICOS (population, intervention, comparator, outcomes, and study design) definition.

From the final selected articles, we extracted the following variables: first author; year of publication; region (country); study design (e.g., observational study or clinical trial); the existence of comparators (e.g., only one arm, or two arms); cancer type and sample size (e.g., metastatic cancer or advanced lung cancer); list of patient-reported outcomes assessed (e.g., patient-reported physical symptoms and/or psychological consequences); instruments used for patient-reported symptom monitoring (e.g., the Edmonton Symptom Assessment System [ESAS]); when (frequency) and how patient-reported symptoms were collected (electronically via a web platform, or using a mobile application [e.g., alert email to the treatment oncologist]); health outcomes measured to evaluate the impact of patient-reported symptom collection; summary of results; main article conclusion; and quality of the study.

Two researchers independently assessed the quality of the studies reviewed against the 30 points of the CONSORT statement for clinical trials [15], the 22 essential points of the STROBE declaration for observational studies [16], and the 24 points of the Consolidated Health Economic Evaluation Reporting Standards (CHEERS) checklist for the reporting of economic evaluations [17]. To unify the costs reported from different studies, costs were converted into Spanish euros (2021) using the CCEMG-EPPI center Cost Converter tool [18].

## 3. Results

### 3.1. General Results

We first identified 1248 studies in MedLine/PubMed and two studies in sources identified via Google Scholar. Of these, 1190 were excluded as they did not provide relevant information for the purpose of the review. We assessed 60 full-text articles for eligibility (Figure 1) and identified 16 publications that discussed the impact of integrating patient-reported symptom monitoring into cancer care. Table 2 shows the summary of the studies included. Appendix A specifies the articles excluded and the reasons for their exclusion: the most common reasons for exclusion were that the publications did not assess the impact of patient-reported symptom monitoring on health outcomes (*n* = 21/*N* = 44) and they focused on the feasibility and utility of patient-reported outcome measures (PROMs) (*n* = 10/*N* = 44).

In short, we identified 16 publications from 13 different studies: three of them published results from the same clinical trial reported by Basch et al. [19,20,21], whereas one economic evaluation [22] was a pre-specified secondary outcome of the Denis et al. [23] trial. All the studies were implemented in Europe and North America (ten of them in the United States and Canada). Of the 16 studies identified, eight had a randomized clinical trial design [19,20,21,23,24,25,26,27], six were observational studies [28,29,30,31,32,33] and two were economic evaluations [22,34]. Of the clinical trials, five reports [20,24,25,26,27] were randomized controlled trials with a high-quality design, whereas two [19,23] were research letters that communicated brief reports of data. Finally, Nipp et al. [21] reported a secondary analysis based on a randomized clinical trial [20]. Among the observational studies, one had a retrospective matched cohort design [28], whereas four used a prospective-cohort design to examine the consequences of implementing a patient-reported symptom strategy in clinical practice [30,31,32,33]. The remaining study compared a patient-reported symptom strategy prospectively with a historical cohort [29]. 

**Table 2 cancers-13-04615-t002:** Summary of the identified studies.

Author and Year of Publication, Country	Type of Study (a), Comparators (b)	Study Aim	Cancer Type; Sample Size	Patient-Reported Symptoms Assessed, Instrument Used and Frequency	Type of Outcome Evaluated	Summary of Results	Main Conclusion	Study Quality
Barbera et al. [28], 2020, Canada	(a) Observational retrospective cohort. (b) Cancer patients exposed (patients must have completed at least one assessment during the study) vs. not exposed to ESAS	To examine the effect of ESAS exposure on cancer patients’ overall survival	Different types of cancer (most prevalent: prostate [17.4%], breast [14.5%] and hematology [13.4%]). *N* = 257,786 (*n* = 128,893 patients exposed to ESAS matched to 128 893 cancer patients not exposed).	ESAS symptoms (pain, tiredness, nausea, depression, anxiety, drowsiness, appetite loss, well-being, and shortness of breath). Collected before a patient’s visit via a touch screen kiosk and discussed during the clinical encounter.	Clinical outcome: probability of survival at 1, 3, and 5 years.	Probability of survival at 1 year: 81.9% vs. 76.4% Probability of survival at 3 years: 68.3% vs. 66.1% Probability of survival at 5 years: 61.9% vs. 61.4% All comparisons *p* < 0.0001	Patients’ exposure to ESAS collection is associated with improved survival in cancer patients	17/22 (STROBE)
Basch et al. [19], 2017, USA (research letter)	(a) Randomized clinical trial. (b) Patients at patient-reported symptom monitoring vs. usual care	To compare overall survival associated with electronic patient-reported symptom monitoring vs. usual care in cancer patients	Cancer patients initiating chemotherapy for solid metastatic tumors (various types). *N* = 766 patients (*n* = 441 intervention arm vs. *n* = 325 control arm).	Twelve symptoms (appetite loss, constipation, cough, diarrhea, dyspnea, dysuria, fatigue, hot flashes, nausea, pain, neuropathy, and vomiting) Symptoms were collected at or between visits via a web-based questionnaire platform (computer-experienced patients) or free standing. computer kiosks (computer-inexperienced patients). The system included email alerts to the treating oncologist	Clinical outcome: median overall survival	Median follow-up: 7 years (IQR, 6.5–7.8) Median overall Survival: 31.2 months (95% CI, 24.5–39.6) vs. 26.0 months (95% CI, 22.1–30.9) (difference, 5.2 months; *p* = 0.03)	Integration of patient-reported symptoms into the routine care of patients with metastatic cancer was associated with increased survival compared with usual care.	-
Denis et al. [23], 2019, France (research letter)	(a) Randomized clinical trial. (b) Patient-reported symptom monitoring vs. usual care	To compare overall survival associated with electronic patient-reported symptom monitoring vs. usual care in cancer patients	Patients with advanced nonprogressive stages IIA to IV lung cancer *N* = 121 patients (*n* = 60 intervention arm vs. *n* = 61 control arm)	Thirteen symptoms (weight, weight variation, appetite loss, weakness, pain, cough, breathlessness, depression, fever, face swelling, lump under skin, voice changing, blood in sputum). Collected weekly in an electronic form between visits. The approach included an alert email to the treating oncologist.	Clinical outcome: median overall survival	Two years of follow-up Median overall survival: 22.5 months vs. 14.9 months (difference, 7.6 months; HR: 0.59 (95% CI, 0.37–0.96); *p* = 0.03)	Symptom monitoring via weekly web-based patient-reported symptom monitoring treatment for lung cancer was associated with increased survival compared with standard imaging surveillance.	-
Basch et al. [20], 2016, USA	(a) Randomized nonblinded, clinical trial. (b) Web-based self-reporting of symptoms vs. usual care.	To test whether systematic web-based collection of patient-reported symptoms during chemotherapy treatment improves HRQoL and survival, quality-adjusted survival, emergency room use, and hospitalization.	Patients with metastatic breast, genitourinary, gynecologic, or lung cancer-initiating chemotherapy. *N* = 766 patients (*n* = 411 intervention arm vs. *n* = 325 in control arm)	Twelve symptoms (appetite loss, constipation, cough, diarrhea, dyspnea, dysuria, fatigue, hot flashes, nausea, pain, neuropathy, and vomiting) Symptoms collected at or between visits via a web-based questionnaire platform (computer-experienced patients) or free-standing computer kiosks (computer-inexperienced patients). The system included email alerts to the treating oncologist.	Patient-reported outcome: percentage of patients with HRQoL clinically meaningful improvement (≥6 points) at six months (EuroQol EQ-5D Index) Use of resources: percentage of patients who visit the emergency room and who were hospitalized. Time receiving active cancer treatment. Clinical outcome: percentage of patients alive at one year	Patient-reported outcome (HRQoL improvement [≥6 points]): 21% vs. 11% (*p* < 0.001) Use of resources and: visits to the emergency room (34% vs. 41%; *p* = 0.02), hospitalizations (45% vs. 49%; *p* = 0.08), time receiving treatment (mean of 8.2 months (range 0–49) vs. 6.3 months (range 0–41), respectively (*p* = 0.002) Clinical outcome: 75% vs. 69% patients alive at one year (difference 6%; *p* = 0.05)	Symptom self-reporting engages patients as active participants and may improve the experience, efficiency, and outcomes of care	30/30 (CONSORT)
Nipp et al. [21], 2019, USA	(a) Randomized controlled trial (secondary analysis). (b) Usual care vs. patient-reported symptom monitoring	To explore whether age moderates the effects of an electronic symptom monitoring intervention on patients’ QoL, health care utilization, and survival outcomes.	Patients with metastatic genitourinary (32.0%), gynecologic (23.1%), or breast cancer (17.7%) initiating chemotherapy. *N* = 766 patients (*n* = 411 intervention arm g vs. *n* = 325 control arm)	Twelve common symptoms (appetite loss, constipation, cough, diarrhea, dyspnea, dysuria, fatigue, hot flashes, nausea, pain, neuropathy, and vomiting) Symptoms collected at or between visits via a web-based questionnaire platform (computer-literate patients) or free-standing computer kiosks (computer illiterate patients). The system included email alerts to the treating oncologist.	Effect of age (patients <70 versus ≥70 years at enrollment) on HRQoL (EuroQol EQ-5D Index); use of resources (hospitalization, emergency room visit) and survival	Patients’ age did not moderate the effects on QoL or time to first hospitalization. Median time to emergency room visit (young patients): 50.73 months for usual care vs. 21.72 months for patient-reported symptoms, *p* = 0.016 Decreased hazard for death for patient-reported symptoms monitoring (HR = 0.76, *p* = 0.011) among younger patients	Among patients with advanced cancer, age moderated the effects of an electronic symptom monitoring intervention on the risk of ER visits and survival	18/30 (CONSORT)
Patel et al. [29], 2019, USA	(a) Observational study. Cancer registry (prospective and retrospective design) (b) Usual care (historical cohort) vs. Lay health worker (LHW)-led symptom screening intervention	To evaluate the effect of a LHW-led symptom screening intervention on satisfaction, self-reported overall and mental health, health care use, total costs, and survival.	Patients with stage 3 or 4 solid tumors or hematologic malignancies who were receiving care in a community oncology practice (most prevalent: gastrointestinal [27.4%], breast [20.5%] and genitourinary [16.7%]). *N* = 288 patients (*n* = 186 intervention arm vs. *n* = 102 control arm)	ESAS symptoms (pain, tiredness, nausea, depression, anxiety, drowsiness, appetite loss, well-being, and shortness of breath). Symptoms were collected by a LHW (weekly for high-risk and monthly for low-risk patients) by telephone. A physician assistant reviewed symptoms daily and notified the oncology provider.	Patient-reported outcome: patients’ satisfaction with care (question no. 18) and self-reported health status (questions no. 23 and 24) of the Medicare Advantage and Prescription Drug Consumer Assessment of Healthcare Providers and Systems. Use of resources: emergency department visits, hospital admissions, (including readmissions and intensive care unit hospitalizations), and the use of hospice services. Median cost per patient (12 months) Clinical outcome: survival during follow-up (12 months)	Patient-reported outcomes: satisfaction improvement (OR = 1.35; 95% CI, 1.08 to 1.63; *p* = 0.002), overall health (OR: 2.23; 95% CI, 1.49 to 3.32; *p* = 0.001), mental or emotional Health (OR = 2.22; 95% CI: 1.46 to 3.38; *p* = 0.001) after 5 months postdiagnosis. Use of resources and: visits to the emergency room (0.61 [SD: 0.98] vs. 0.92 [SD: 1.53]; *p* = 0.03), hospitalizations (0.72 [SD: 0.96] vs. 1.02 [SD:1.44]; *p* = 0.03). Median total costs: €13,915.0 (IQR: €5301.3–€31,342.1) intervention vs. €20,903 (€10,308.2–€37,537.9) usual care (*p* = 0.01). Clinical outcome: 39% intervention vs. 28% usual care patients had died during 12-month follow-up (HR = 1.21; 95% CI, 0.78 to 1.87; *p* = 0.86)	The LHW-led symptom screening intervention among patients with advanced stages of cancer was associated with improved patient-reported outcomes, no differences in survival, and reduced acute care use and total health care costs.	18/22 (STROBE)
De Raaf et al. [24], 2013, The Netherlands	(a) Randomized non-blinded controlled trial (b) Protocolized patient-tailored treatment (PPT) of physical symptoms vs. care as usual (CAU)	To investigate whether monitoring of physical symptoms coordinated by a nurse has a more favorable effect on fatigue severity than the symptom management included in the standard oncologic care of patients with advanced cancer.	Patients with solid malignancies in palliative care and fatigued Most prevalent: breast (36.8%), gastrointestinal (30.9%) and urogenital cancer (15.8%) *N* = 152 patients (*n* = 76 in the PPT vs. *n* = 76 in CAU)	Nine symptoms (pain, nausea, vomiting, constipation, diarrhea, lack of appetite, shortness of breath, cough, and dry mouth) Collected during meetings with the nurse specialist at the outpatient clinic. When patients rated a certain symptom ≥ 4 (from 0 to 10), the nurses asked the oncologist to start an appropriate treatment using specific protocols.	Patient-reported outcome: fatigue (MFI scores at T1 = 1 month; T2 = 2 months; T3 = 3 months after random assignment). Influence of fatigue on daily life (BFI-I). Anxiety and depressed mood (HADS).	Patient-reported outcome: MFI scores: T1 (mean difference, −0.84 [SE: 0.31]; *p* = 0.007), T2 (mean difference, −1.14 [SE: 0.40]; *p* = 0.005), T3 (mean difference, −0.90 [SE, 0.50]; *p* = 0.07). Interference of fatigue with daily life: the PPT group reported a decrease in the interference of fatigue with daily life (maximal effect size, 0.64; *p* < 0.001) Anxiety: anxiety decreased in the PPT group as compared with the CAU group (maximal effect size, 0.32; *p* = 0.001	In fatigued patients with advanced cancer, nurse-led monitoring and protocolized treatment of physical symptoms are effective in alleviating fatigue	29/30 (CONSORT)
Diplock et al. [30], 2019, Canada	(a) Prospective observational study (b) Cancer patients screened prior to and after ESAS implementation	To assess the impact of implementing ESAS screening on HRQoL and patient satisfaction with care in ambulatory oncology patients.	Ambulatory oncology patients. Most prevalent: breast (25.4%), hematologic oncology (16.8%) and head and neck cancer (16.4%) *N* = 268 patients (*n* = 160 prior to ESAS site implementation vs. *n* = 108 after ESAS implementation)	ESAS symptoms (pain, tiredness, nausea, depression, anxiety, drowsiness, appetite loss, well-being, and shortness of breath). ESAS was collected via a touch screen kiosk (at baseline (T1), and two weeks later (T2))	Patient-reported outcome: HRQoL (EORTC-QLQ-C30). Patients’ satisfaction with care (PMH/PSQ-MD-24 and EORTC-OUTPATSAT35 RT and CT)	No significant differences between HRQoL and satisfaction outcomes from the matched non-ESAS and ESAS groups (at T1 and T2) Nausea and Vomiting punctuations significantly decreased over time: mean 10.75 (SD: 19.40) prior to ESAS, mean 7.44 (SD: 12.96) at T1 and mean 8.13 (SD: 14.64) at T2 Constipation: mean 28.08 (SD: 32.23) prior to ESAS, mean 13.28 (SD: 21.51) at T1 and mean 10.60 (SD: 18.50) at T2	There was no impact of early-ESAS screening on HRQoL or satisfaction outcomes	16/22 (STROBE)
Baratelli et al. [31], 2019, Italy	(a) Observational, prospective cohort (b) Usual care vs. patient-reported symptoms	To compare two groups of patients: a first group, visited using the usual modality of toxicity and symptoms collection and management, and a second group, visited after the introduction of patient-based assessment of symptoms and toxicities in routine clinical practice	Cancer patients receiving active anti-cancer treatment as outpatients. Most prevalent colorectal cancer (32.7%), lung cancer (19.9%), and pancreatic cancer (14.7%) *N* = 211 (*n* = 119 usual care vs. *n* = 92 patient-reported symptoms patients)	Thirteen symptoms (mouth problems, nausea, vomiting, constipation, diarrhea, dyspnea, skin problems, nail problems, itching, hand/foot problems, fatigue, pain, and other issues) Symptoms collected by a nurse before each visit and were paper based. They were delivered to the physician, who could consult them before the visit.	Patient-reported outcome: HRQoL (mean change from baseline to 1 month of (EORTC-QLQ-C30) scores)	Patient-reported outcome: mean change from baseline of global QoL was −1.68 (SE: 1.88) for usual care vs. 2.54 (SE: 2.32) in patient-reported symptoms (*p* = 0.004).	Introduction of patient-reported symptom monitoring in clinical practice produced a significant QoL improvement compared to the traditional modality of visit.	17/22 (STROBE)
Strasser et al. [25] 2016, Switzerland	(a) Multicenter cluster randomized clinical trial (b) Patient-reported symptoms and monitoring vs. usual care	To test the effects of the E-MOSAIC intervention (patient-reported symptoms and monitoring) in patients with incurable cancer getting a new line of chemotherapy with palliative intent.	Patients received anticancer treatment with palliative intent Most prevalent non-small-cell lung cancer. (18.9%), colorectal cancer (14.4%) and breast cancer (10.2%) *N* = 264 (*n*= 119 usual care vs. *n* = 145 patient-reported symptoms patients)	ESAS symptoms, and three additional outcomes (estimated nutritional intake, body weight change, and Karnofsky performance) Electronic patient-reported symptoms collected weekly. In the intervention group, longitudinal monitoring sheet was printed and was immediately given to the oncologists.	Patient-reported outcome: HRQoL (mean change from baseline to 6 weeks of (G-QoL) scores for EORTC-QLQ-C30; difference between the arms)	Patient-reported outcome: the difference in HRQoL between arms was 6.84 (−1.65, 15.33) (*p* = 0.1) in favor of the intervention arm	Monitoring of patient symptoms, clinical syndromes, and their management clearly reduced patients’ symptoms, and QoL (although the difference was not statistically significant)	30/30 (CONSORT)
Kneuertz et al. [32], 2020, USA	(a) Observational prospective pilot study (b) One arm	To understand the utility of a mobile application platform to engage patients whilst gathering data on patient compliance, perioperative experience and satisfaction.	Patients diagnosed with lung cancer who were scheduled for robotic surgery *N* = 50 patients	Post-discharge recovery assessment of pain, anxiety and mood. Social role functioning and return to work Collected daily (pain, anxiety and mood) and at days 14 and 30 post-discharge (social role functioning and return to work) through a mobile application. The care team had access to patients’ reports.	Patient-reported outcome: satisfaction with their hospital stay (ad-hoc questionnaire)	Patient-reported outcome: 77.4% of patients gave the highest-ranking (“excellent”) for the care received, and 93.5% reported they would recommend the hospital to others based on their experience	A mobile device platform may serve as an effective mechanism to record perioperative patient-reported symptoms and satisfaction while facilitating patient-provider engagement in perioperative care.	17/22 (STROBE)
Riis et al. [26], 2020, Denmark	(a) Pilot randomized controlled trial. (b) standard follow-up care vs. individualized follow-up care	To evaluate the patients’ satisfaction with the care provided when using electronic patient-reported symptom monitoring to individualize follow-up care in women with early breast cancer receiving adjuvant endocrine therapy.	Postmenopausal women with early breast cancer receiving adjuvant endocrine therapy. *N* = 134 (*n* = 64 standard follow-up care vs. *n* = 60 individualized follow-up care)	Quality of life (EORTC QLQ-C30), including three symptom scales (fatigue, nausea and vomiting, and pain), and six single items (appetite loss, diarrhea, dyspnea, constipation, insomnia, financial impact) Collected every third month over a two-year period electronically. The principal investigator monitored incoming questionnaires.	Patient-reported outcomes: satisfaction with the care provided as measured by four items from the PEQ. Secondary outcomes were use of consultations and adherence to treatment (collected every third month over a two-year period.	Satisfaction with follow-up care and adherence: no statistically significant differences between standard follow-up care vs. individualized follow-up care. Use of resources: patients in standard care attended 4.3: (CI 3.9–4.7) consultations each vs. 2.1 (CI: 1.6–2.6) in patients attending individualized care (*p* < 0.001).	A significant reduction in consultations was observed for the group attending individualized care without compromising the patients’ satisfaction, quality of life, or adherence to treatment	24/30 (CONSORT)
Nipp et al. [27], 2019, USA	(a) Nonblinded, pilot randomized controlled trial. (b) Usual care vs. patient-reported symptom monitoring.	To assess the feasibility and preliminary efficacy of symptom monitoring for improving symptom burden and health care utilization among hospitalized patients with advanced cancer.	Hospitalized patients with advanced cancer admitted to oncology service. Most prevalent: gastrointestinal (36.6%), lung (22%) and head and neck cancer (10%) *N* = 150 patients (*n* = 75 in the patient-reported symptoms monitoring vs. *n* = 75 in usual care arm)	Physical symptoms (pain, fatigue, drowsiness, nausea, appetite loss, dyspnea, constipation, and diarrhea) Psychological symptoms (depression, anxiety, and well-being) Collected daily using tablet computers. Study staff presented patients’ reports each day to the clinical staff.	Use of resources and cost outcomes: hospital length of stay, time to first unplanned readmission within 30 days.	No significant difference in patients’ hospital length of stay (B = 0.16, 95% CI: −1.67–1.99; *p* = 0.862). Patients assigned to symptom monitoring had a lower risk of readmissions (HR = 0.68, 95% CI: 0.37–1.26; *p* = 0.224).	Intervention patients had lower readmission risk, although this difference was not significant.	27/30 (CONSORT)
Howell et al. [33], 2020, Canada	(a) Observational prospective study (b) Usual care vs. patient-reported symptom monitoring	To determine if there was a difference in relative rates for emergency department visits and hospitalizations for the iPEHOC (Improving Patient Experience and Health Outcomes Collaborative intervention) exposed population compared with contemporaneous controls	Different types of cancer. Most prevalent: prostate cancer (16%), breast cancer (14.9%), and gynecological cancer (8.7%) *N* = 129,797 (pre-intervention *n* = 70,854 and intervention group *n* = 58,943)	The iPEHOC intervention included patient-reported symptoms collection through an electronic system in addition to clinicians’ and patients’ educational interventions. ESAS symptoms (pain, tiredness, nausea, depression, anxiety, drowsiness, appetite loss, well-being, and shortness of breath) In addition, Brief Pain Inventory, the Cancer Fatigue Scale, the Generalized Anxiety Disorder, and the PHQ-9	Use of resources and cost outcomes: emergency department visits, hospitalization rates and drug prescriptions for each clinic (expressed as difference in difference (DID) approach).	Use of resources: DID = −0.223 in the RR for emergency department visits for the intervention compared with controls over time (0.947, CI 0.900–0.996). There was also lower DID in palliative care visits (−0.0097), psychosocial oncology visits (−0.0248) and antidepressant prescriptions in the exposed population compared with controls.	Facilitating uptake of patient-reported symptoms data may impact healthcare utilization.	19/22 (STROBE)
Lizee et al. [22] 2019, France	(a) Economic evaluation based on the data from a multicenter randomized clinical trial (b) Usual care vs. patient-reported symptom monitoring	To assess and compare the overall cost of surveillance in the patient-reported symptom monitoring and control arms. To assess the cost-effectiveness of this surveillance based on web-based patient-reported symptom monitoring, compared to conventional surveillance.	Patients with advanced non-progressive stages IIA to IV lung cancer *N* = 121 patients (*n* = 60 in the patient-reported symptoms monitoring vs. *n* = 61 in the usual care arm)	Thirteen symptoms (weight, weight variation, appetite loss, weakness, pain, cough, breathlessness, depression, fever, face swelling, lump under skin, voice changing, blood in sputum). Collected weekly by patients in an electronic form. The PRO system automatically triggered an alert email to the Treating.	Use of resources and cost outcomes: average annual cost per patient, including the following use of resources (consultation, imaging, trip, conventional follow-up (including the e-PRO system-related costs in the experimental arm)) Cost–utility analysis (incremental cost-effectiveness ratio per life-year gained and per QALY)	Use of resources: average annual cost of surveillance follow-up was (€2828.1/year/patient) compared to control (€1129.2/year/patient). The patient-reported symptom monitoring approach presented an incremental cost-effectiveness ratio of €10,500.9 per life-year gained and €18,107.9 per QALY (cost-effective)	Surveillance of lung cancer patients using web-based patient-reported symptom monitoring reduced the follow-up costs and represented a cost-effective strategy.	23/24 (CHEERS)
Nixon et al. [34] 2018, Canada	(a) Economic evaluation based on the data from a randomized nonblinded, clinical trial (b) Usual care vs. patient-reported symptom monitoring	To evaluate the cost-effectiveness of a patient reported outcome tool for symptom monitoring in patients undergoing treatment for advanced or metastatic cancer compared to standard of care symptom monitoring from the perspective of the public payer in Alberta.	Patients with metastatic solid tumors receiving systemic therapy	Twelve symptoms (appetite loss, constipation, cough, diarrhea, dyspnea, dysuria, fatigue, hot flashes, nausea, pain, neuropathy, and vomiting) Symptoms collected at or between visits via a web-based questionnaire platform (computer-experienced patients) or free-standing computer kiosks (computer-inexperienced patients). The system included email alerts to the treating oncologist.	Cost–utility analysis (incremental cost-effectiveness ratio per life-year gained and per QALY)	The patient-reported symptom monitoring approach presented an incremental cost-effectiveness ratio of €7575.9 per QALY (cost-effective)	The use of a PRO tool for symptom monitoring yields a cost per QALY of €7575.9 that would be considered a good value for money at the typically accepted Canadian standard of $50,000 (€28,163.2) per QALY	21/24 (CHEERS)

Abbreviations: BFI-I (Brief Fatigue Inventory); CAU (care as usual); ESAS (Edmonton Symptom Assessment System); EORTC-OUTPATSAT35 RT and CT (The European Organization for Research and Treatment of Cancer Outpatient Satisfaction with Radiation and Chemotherapy); EORTC-QLQ-C30 (European Organisation for Research and Treatment of Cancer Quality of Life Questionnaire C-30); G-QoL (Global Quality of Life Scale); HADS (Hospital Anxiety and Depression Scale); HRQoL (Health-Related Quality of Life); IQR (interquartile range); MFI (Multidimensional Fatigue Inventory); LHW (Lay health worker); PeQ (patient experience questionnaire); PPT (patient-tailored treatment of physical symptoms); QALY (Quality-Adjusted Life Year).

All the studies except for one [32] examined the impact of patient-reported symptom monitoring compared to usual care monitoring. In the studies reviewed, usual care typically involved patients and their oncologists discussing symptoms during scheduled visits, whereas patient-reported symptom monitoring involved patients regularly self-reporting cancer-related symptoms. In most cases (*n* = 13), patient-reported symptoms were collected electronically through websites, mobile applications, touch screen kiosks, or tablets during and between scheduled visits [19,20,21,22,23,25,26,27,28,30,32,33,34]. 

In six studies [19,20,21,22,23,34], the approach included email alerts sent to the treating physician when patient-reported symptoms matched predefined criteria for severity. These alerts enabled clinicians to take rapid clinical actions in response to these reports.

The most frequently monitored physical cancer-related symptoms were pain (*n* = 16), loss of appetite (*n* = 14), and nausea (*n* = 13), whereas only six out of the 16 studies assessed depression symptoms. 

Twelve studies addressed patients with different types of cancers [19,20,21,24,25,27,28,29,30,31,33,34], and eight of them targeted advanced or metastatic cancer patients [19,20,21,24,25,27,29,34]. Four publications included specific types of cancers [22,23,26,32], most of which evaluated the impact of patient-reported symptom monitoring on lung cancer patients [22,23,32]. 

### 3.2. Impact of Patient-Reported Symptoms on Health Outcomes

Of the 16 studies identified, six [19,20,21,23,28,29] assessed the impact of the integration of patient-reported symptom monitoring on survival; nine [20,21,24,25,26,29,30,31,32] evaluated the impact of patient-reported symptom monitoring on other patient-reported outcomes such as HRQoL and satisfaction with treatment, and eight [20,21,22,26,27,29,33,34] assessed economic outcomes, for example, the use of resources and costs of a patient-reported symptom monitoring strategy versus usual symptom monitoring.

#### 3.2.1. Overall Survival

Six out of the 16 publications assessed the effects of patient-reported symptom monitoring on cancer patients’ overall survival [19,20,21,23,28,29]. One of them [21] was a secondary analysis of data to explore the effect of age on different outcomes, including survival. All the studies, except for one registry [29], showed that patient-reported symptom surveillance led to significantly improved survival compared to usual care symptom monitoring. The clinical trial reported by Basch et al. [20], which included patients initiating chemotherapy for solid metastatic tumors (*N* = 766)—mainly prostate (17.4%), breast (14.5%), and hematologic neoplasms (13.4%)—reported an overall survival benefit of 6% (percentage of patients alive: 75% for patient-reported symptom monitoring vs. 69% for usual care; *p* = 0.05) at one year of follow-up. Long-term results of the same trial [19] showed a gain of 5.2 months in favor of the patient-reported symptom monitoring arm in a median follow-up of seven years (median overall survival: 31.2 months (95% CI, 24.5–39.6) vs. 26.0 months (95% CI, 22.1–30.9) for usual care; *p* = 0.03). Additionally, Nipp et al. [21] conducted a secondary analysis on data from Basch et al. [19,20]. They found that the survival benefits associated with patient-reported symptom surveillance were limited to younger patients (<70 years): thus, the authors recorded a significant decrease in the hazard for death for patient-reported symptom monitoring (HR = 0.76; *p* = 0.011) among younger patients (<70 years); however, they did not find significant survival benefits for older patients (≥70 years) assigned to the intervention arm (HR = 1.06; *p* = 0.753).

Denis et al. [23] reported an increase of almost eight months’ survival, after two years of follow-up, for the patient-reported monitoring arm (median overall survival: 22.5 months vs. 14.9 months usual care; HR: 0.59 (95% CI, 0.37–0.96); *p* = 0.03) in a cohort of lung cancer patients (*N* = 121) with advanced nonprogressive disease (stages IIA to IV). In turn, the study of Barbera et al. [28] reported an absolute overall survival benefit at one year of 5.5% for patients exposed to patient-reported symptom monitoring vs. usual care (survival probability of 81.9% vs. 76.4%; *p* < 0.0001) after one-year follow-up. 

#### 3.2.2. Health-Related Quality of Life (HRQoL)

Five studies evaluated the impact of patient-reported symptom surveillance on HRQoL [20,21,25,30,31]. The questionnaires used to assess HRQoL were the generic EuroQol 5-Dimensional Scale (EQ-5D) [20,21] or the cancer-specific Organization for Research and Treatment of The Cancer Quality of Life Questionnaire (EORTC-QLQ-C30) [25,30,31]. Most of the studies demonstrated how the interventions improved HRQoL compared to usual symptom monitoring. In the study by Basch et al. [20] in metastatic patients, the percentage of patients who registered a clinically meaningful HRQoL improvement (≥6 points) after six months was significantly higher in the patient-reported symptoms arm than in the usual care arm (21% vs. 11%; *p* < 0.001). A secondary analysis [21] showed that the HRQoL benefit obtained from electronic symptom monitoring did not change as a function of age (B = −0.02; *p* = 0.994). Baratelli et al. [31] demonstrated that the introduction of patient-reported symptoms collection led to a significant benefit in HRQoL in patients actively receiving anti-cancer treatment (mean global QoL change from baseline to one month of follow-up: −1.68 (SE: 1.88) for usual care vs. 2.54 (SE: 2.32) for patient-reported symptoms (*p* = 0.004)). 

Similarly, Strasser et al. [25] determined a between-arm difference in HRQoL of 6.84 (−1.65, 15.33) (*p* = 0.1) in favor of the intervention arm for a cohort of cancer patients in palliative care (*N* = 264).

In turn, Diplock et al. [30] observed that HRQoL remained unchanged two weeks after the implementation of a patient-reported symptom monitoring strategy on ambulatory oncology patients (*N* = 268) mainly diagnosed with breast (25.4%), hematologic (16.8%), and head and neck cancer (16.4%). However, nausea, vomiting and constipation scores significantly decreased after patient-reported symptom monitoring: for nausea and vomiting, the mean was 7.44 (SD: 12.96) at baseline, and 8.13 (SD: 14.64) two weeks after patient-reported symptom monitoring; for constipation, the mean was 13.28 (SD: 21.51) at baseline, and 10.60 (SD: 18.50) two weeks after patient-reported symptom monitoring.

#### 3.2.3. Patient-Reported Satisfaction

Four studies rated patients’ levels of satisfaction with care [26,29,30,32]. One of them used an ad-hoc questionnaire [32] for this purpose. In contrast, Diplock et al. [30] applied the Princess Margaret Hospital Patient Satisfaction Questionnaire (PMH/PSQ-MD-24), and Riis et al. [26] employed the Patient Experience Questionnaire (PEQ) to assess satisfaction in postmenopausal women with early breast cancer. Patel et al. [29] assessed patients’ satisfaction with care using the “satisfaction with provider” item of the validated Medicare Advantage and Prescription Drug Consumer Assessment of Healthcare Providers and Systems. As a result, two reports showed that although patients were satisfied with the care received, non-statistically significant differences were observed when comparing patient-reported symptom surveillance with usual care [26,30]. In contrast, the study of Patel et al. [29] demonstrated that compared with usual care, the introduction of an LHW-led symptom screening intervention involved a significant improvement in satisfaction with care (OR = 1.35; 95% CI, 1.08 to 1.63; *p* = 0.002) 

#### 3.2.4. Patient-Reported Fatigue

One study aimed at palliative cancer patients found out how monitoring and protocolizing the management of patient-reported symptoms alleviated fatigue in cancer patients in palliative care (*N* = 152) [24]. Specifically, Multidimensional Fatigue Inventory (MFI) scores were significantly lower in the patient-reported symptom monitoring arm than those reported in usual care after one and two months of follow-up (mean difference, −0.84 [SE: 0.31]; *p* = 0.007 after one month and mean difference, −1.14 [SE: 0.40]; *p* = 0.005 after two months). Additionally, patients in the patient-reported symptom monitoring group had a significant decrease in the interference of fatigue with daily life (maximal effect size, 0.64; *p* < 0.001).

#### 3.2.5. Adherence

Of the 16 studies, only one dealt with the effects of patient-reported symptom follow-up on medication adherence in early breast cancer patients receiving adjuvant endocrine therapy (*N* = 134) [26]. However, no statistically significant differences were observed when comparing usual symptom follow-up care. 

#### 3.2.6. Economic Outcomes

Eight publications assessed the use of resources and economic impact of a patient-reported symptom monitoring [20,21,22,26,27,29,33,34], mainly in advanced or metastatic cancer [20,21,22,27,29,34]. 

They showed that patient-reported symptom monitoring led to a substantial decrease in healthcare usage compared to usual care. Visits to the emergency room and hospitalizations and readmissions were lower in the intervention group than in the usual care monitoring group. In the study by Basch et al. [20], the percentage of metastatic cancer patients that visited the emergency room and were hospitalized was lower in the patient-centered surveillance arm than in the usual care arm (34% vs. 41%, *p* = 0.02; 45% vs. 49%, *p* = 0.08, respectively), although differences in hospitalized patients were not significant. A secondary analysis of this clinical trial according to patients’ age [21] showed that the median time spent in the emergency room was significantly lower for the patient-reported symptoms arm in younger patients compared to the control arm (21.72 months vs. 50.73 months; *p* = 0.016). 

Riis et al. [26] reported fewer consultations per patient by patients attending individualized care compared to standard care (2.1 [CI: 1.6–2.6] vs. 4.3 [CI 3.9–4.7]; *p* < 0.001). In another report [27] focusing on hospitalized patients with advanced cancer (*N* = 150), patients assigned to symptom monitoring had a lower risk of readmissions (HR = 0.68, 95% CI: 0.37–1.26; *p* = 0.224). Howell at al. [33], who assessed differences in healthcare utilization in a large cohort of cancer patients (*N* = 129 797) who were mainly diagnosed with prostate cancer (16%), breast cancer (14.9%), and gynecological cancer (8.7%), showed fewer emergency department visits (difference in difference (DID) = −0.223), palliative care visits (DID = −0.0097), psychosocial oncology visits (DID = −0.0248) for the intervention group compared with control subjects over time. In turn, an LHW-led symptom screening intervention [29] was associated with an approximately 30% reduction in emergency visits (0.61 [SD: 0.98] vs. 0.92 [SD: 1.53]; *p* = 0.03) and hospitalizations (0.72 (SD: 0.96) vs. 1.02 (SD: 1.44); *p* = 0.03) than the standard of care. In addition, median total costs (including all inpatient and outpatient services for 12 months) were also significantly lower in the intervention group than in the usual symptom monitoring cohort (€13,915.0 [IQR: 5301.3–31,342.1] vs. €20,903 [€10,308.2–€37,537.9]; *p* = 0.01). In contrast, an economic evaluation [22] showed that the average annual cost was higher for the patient-reported symptom monitoring arm (average annual cost of €2828.1/year/patient) compared to controls (€1129.2/year/patient). However, when including its survival benefits [23], patient-reported symptom monitoring was found to be a cost-effective strategy (incremental cost-effectiveness ratio of €10,500.9 per life-year gained and €18 107.9 per Quality-Adjusted Life Year [QALY]) from the French national health insurance perspective. Another cost-effectiveness analysis based on data from Basch et al. [20] trial in metastatic patients, demonstrated that the electronic patient-reported symptom monitoring strategy was a cost-effective alternative (incremental cost-effectiveness ratio of €10,500.9 per QALY) from the perspective of the public payer in Canada. 

## 4. Discussion

Our systematic literature review shows that patient-reported symptom monitoring—mainly electronic-based—provides substantial benefits in terms of clinical (overall survival), other patient-reported (such as HRQoL and satisfaction, etc.), and economic (use of healthcare resources, costs, cost-effectiveness, etc.) outcomes. 

On the one hand, we found that active patient-reported monitoring is associated with increased survival equal to or greater than five months compared with usual care in advanced cancer. One hypothesis for this remarkable benefit is that patient-reported surveillance allows clinicians to both respond earlier to worsening symptoms and tumor recurrence through the email alerts [19,20,21,23]; thus, clinicians can perform rapid interventions that prevent complications, unexpected hospitalizations, or chemotherapy withdrawal. Indeed, improvements in symptom management may also allow patients to tolerate and, consequently, to benefit from chemotherapy for a longer time than when in usual care—for example, in the study conducted by Basch et al. [20], the mean treatment time in the intervention group was 8.2 months vs. 6.3 months in the usual care group; *p* = 0.002.

Although survival benefits were robust and consistent across most of the studies, we acknowledge that extrapolating these findings to the general population may be risky. Firstly, most studies that demonstrated this benefit were placed in a single country and department [19,20,21]. Added to this, most of them were randomized controlled clinical trials [19,20,21,23], where patients’ eligibility criteria were very strict and the interventions very well-defined. Moreover, the results from a secondary analysis by Basch et al. [21] pointed out that survival benefits might be restricted to younger patients (<70 years). The authors speculated that for elderly patients, many other factors apart from symptom monitoring might influence survival, such as mobility, cognitive function, and the availability of social support. Additionally, elderly patients are more likely to be computer illiterate, which could compromise the effectiveness of this type of intervention. Finally, another point to consider is that the effect of patient-reported symptom monitoring on survival seems to be stronger in the first year after diagnosis or recurrence than in the following years. Accordingly, Barbera et al. [28] observed an absolute overall survival benefit of 5.5% the first year, whereas it was 2.2% after 3 years and 0.5% after 5 years since diagnosis or recurrence. Thus, the authors hypothesized that early symptom identification and treatment imply a more noticeable benefit to patients when their general health state is worse, which generally coincides with the emergence and the recurrence of the disease.

On the other hand, patient-reported symptom monitoring significantly enhanced different patient-reported outcomes compared to usual care. For instance, some studies observed an HRQoL improvement associated with this monitoring strategy. The main reason for the HRQoL improvement might be the reduction in the severity of the symptoms. For example, better pain control might result in a better trend in global HRQoL [31]. Only one of the studies [30] showed that HRQoL did not differ between patient-reported and usual monitoring. The absence of differences may be due to low sensitivity for detecting HRQoL changes in a short time period (two weeks) and limited statistical power due to the small sample size. Despite this limitation, the authors still reported a significant improvement in some symptoms, reflecting greater attention to physical symptoms after patient-reported symptom implementation. Additionally, most of the patients surveyed were highly satisfied with the patient-reported symptoms approach [26,29,30,32], although no significant differences were found when comparing this monitoring strategy with usual care in two of the studies [26,30]. Despite not being significant, these results can be considered a positive outcome: they indicate that more individualized patient-reported monitoring does not compromise patients’ satisfaction with care. 

Finally, most studies showed that patient-reported symptom monitoring led to economic benefits—namely, substantial savings on healthcare use of resources—compared to usual care symptom follow-up [20,21,26,27,33]. These savings may be attributable mainly to better symptom control, which prevents unplanned emergency room visits [20,27,29,33] and hospital readmissions [27,29]. 

With respect to medical costs, one observational study [29] demonstrated how an LHW-led symptom screening intervention significantly decreased healthcare costs, mainly due to reduced acute care. In contrast, Lizee et al. [22] showed that patient-reported symptom monitoring increased the average annual cost of surveillance, mainly due to the extra expenses associated with the electronic system. However, when adding the survival benefits associated with patient-reported symptom monitoring [23], the intervention proved to be cost-effective from the French national health insurance perspective. Despite this positive result, some study limitations must be taken into consideration: (1) the analysis was limited to the time horizon of the trial (two years follow-up); (2) the EQ-5D utilities were not measured in the trial and were derived from prior research; (3) the study was conducted from a payer’s perspective (the authors did not consider societal costs [e.g., productivity costs or informal care]). 

### 4.1. Other Implications of Patient-Reported Symptom Monitoring

Besides the benefits of survival, HRQoL, and economic outcomes already reported, monitoring patient-reported symptoms might also have other implications in clinical practice that deserve particular attention despite being beyond the scope of the present review. For instance, this systematic monitoring might make cancer patients more aware of their symptoms and disease as a whole [35,36,37]. Thus, patients can better inform their providers about their health status [37] and feel more comfortable initiating discussions with the medical team about their symptoms or other concerns [38,39]. Therefore, it seems that the systematic collection of patient-reported symptoms helps to build a better rapport and break down barriers in communication between patients and clinicians [40]. Better patient–physician communication eventually helps physicians to identify health problems that might otherwise go unnoticed [22,39], better focus the consultation, and devote more time to patients’ main concerns [41]. Another interesting implication involves the increasing importance of telemedicine and electronic devices and software in current clinical practice, especially after the outbreak of the crisis caused by severe acute respiratory syndrome coronavirus 2 (SARS-CoV-2). In this regard, remote monitoring of patient-reported symptoms provides an excellent opportunity to improve cancer management, reduce contact between individuals, and prevent unnecessary visits to hospital. 

Despite the well-documented benefits associated with patient-reported symptom monitoring, significant limitations might still prevent its widespread implementation in clinical practice. The major barrier might be logistical, as many patients are unfamiliar with electronic devices or are computer illiterate, especially elderly patients. Thus, these groups of patients are at higher risk of being excluded from this technology. The second barrier is related to patients’ compliance as they must be engaged to fill in the questionnaires regularly (sometimes every week). As a result, compliance might decrease over time. Lastly, monitoring of patient-reported symptoms may be a burden for care providers as it may increase care costs. For this reason, more cost-effectiveness studies should be conducted in different settings to demonstrate how the clinical benefits outweigh the possible extra costs associated with patient-reported active surveillance. 

### 4.2. Study Limitations 

Finally, some limitations of the present study should be considered. The first is related to its design. A systematic review examines and synthesizes the information on a subject available in the literature and includes some bias associated with the publications. The main biases associated with publications have already been considered throughout the discussion section. The second limitation concerns the research source and languages of publication as we have limited the search to one database and to publications in English and Spanish. We acknowledge that these restrictions may result in the omission of patient-reported symptom monitoring interventions discussed in other databases or languages. 

Although the benefits derived from patient-reported symptom surveillance are well-documented and consistent across the different studies, a formal meta-analysis might be necessary to validate the effect of the symptom monitoring intervention on the different outcomes evaluated. 

## 5. Conclusions

In this review, we have appraised recent evidence on the benefits of patient-reported symptom monitoring in cancer patients. The evidence illustrates that this monitoring modality entails an outstanding survival benefit for advanced cancer patients—five or more additional survival months than for usual care monitoring. Additionally, patient-reported symptom monitoring showed a positive effect on patients’ HRQoL and satisfaction with care. It led to a substantial decrease in healthcare usage, preventing unplanned emergency room visits and hospital readmissions. Monitoring patient-reported symptoms might also have other implications in clinical practice that could be the subject of another study, such as promoting patients’ disease awareness or improving patient–physician communication and relations. 

Notwithstanding these advantages, there are still logistical barriers that prevent its widespread implementation—especially in its electronic modality. In addition, the real-world effectiveness and the cost-effectiveness of this strategy are yet to be proven in different settings.

## Figures and Tables

**Figure 1 cancers-13-04615-f001:**
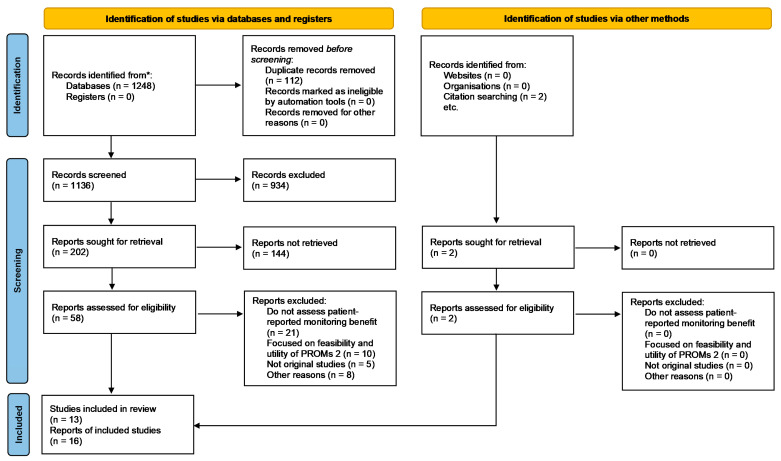
PRISMA flow diagram. * MedLine/PubMed database.

**Table 1 cancers-13-04615-t001:** Eligibility criteria defined by PICOS.

Inclusion Criteria	
Population	Cancer patients (any type and stage)
Intervention	Patient-reported symptom monitoring, regardless of the modality: paper-based or electronic
Comparator	-
Outcome	Health results as clinical (e.g., survival), patient-reported (e.g., HRQoL, general perception or feelings of well-being, satisfaction), or economic (use of healthcare resources, costs, cost-effectiveness) outcomes
Study Design	Original articles

Abbreviations: HRQoL (Health-Related Quality of life).

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
