# Peer review of "Additional Value of Patient-Reported Symptom Monitoring in Cancer Care: A Systematic Review of the Literature"

_cancers, 2021, doi:10.3390/cancers13184615_

Round 1

Reviewer 1 Report

The revised version gained significant improvement in data presentation and discussion of the results.

This manuscript is a resubmission of an earlier submission. The following is a list of the peer review reports and author responses from that submission.

Round 1

Reviewer 1 Report

This is a very nice and well done review focussing towards patient reported outcome. Few minor comments should be considered:

chapter 3.2.1: the data should be visualized and the survival benefit needs to be related to the entities and more in details to cohort characteristics

chapter 3.2.6: for cost analysis the search criteria are likely not appropriate, though additional studies investigating such topics might be missed. This should be discussed

Chapter 4: the discussion is somewhat speculative and, in part, redundant to chapter 3. The most important weeakness is the missing discussion of the limitations of the included trials. The authors limit this point to their own limitations of the review generation, but critical reflection of the included reports is mandatory.

Reviewer 2 Report

This is a well-conducted systematic review w/o meta-analysis to assess the additional value of patient-reported symptom monitoring in

cancer care.

The topic is interesting and the systematic review process seems reasonable.

However, the following points should be amended before the publication.

Major comments.

This systematic review clarified that patient-reported symptoms monitoring

1)entailed overall survival by 5 months or more.

2)improved patients' HRQoL

3)decreased healthcare usage

These beneficial effects should be assessed numerically.

I strongly advice authors to add meta-analysis focusing on RCTs to make this review more persuasive.

This is the current standard of systematic review.

Besides, Prisma Checklist should be presented.

Minor comments

None

Round 2

Reviewer 2 Report

Unfortunately, the manuscript was not sufficiently chagned.